# Persistence and *In Vivo* Evolution of Vaginal Bacterial Strains over a Multiyear Time Period

Michael France,[a,b] Bing Ma,[a,b] Jacques Ravel[a,b]

[a]Institute for Genome Sciences, University of Maryland School of Medicine, Baltimore, Maryland, USA
[b]Department of Microbiology and Immunology, University of Maryland School of Medicine, Baltimore, Maryland, USA

**ABSTRACT** It is not clear whether the bacterial strains that comprise our microbiota are mostly long-term colonizers or transient residents. Studies have demonstrated decades-long persistence of bacterial strains within the gut, but persistence at other body sites has yet to be determined. The vaginal microbiota (VMB) is often dominated by *Lactobacillus*, although it is also commonly comprised of a more diverse set of other facultative and obligate anaerobes. Longitudinal studies have demonstrated that these communities can be stable over several menstrual cycles or can fluctuate temporally in species composition. We sought to determine whether the bacterial strains that comprise the VMB were capable of persisting over longer time periods. We performed shotgun metagenomics on paired samples from 10 participants collected 1 and 2 years apart. The resulting sequences were *de novo* assembled and binned into high-quality metagenome assembled genomes. Persistent strains were identified based on the sequence similarity between the genomes present at the two time points and were found in the VMB of six of the participants, three of which had multiple persistent strains. The VMB of the remaining four participants was similar in species composition at the two time points but was comprised of different strains. For the persistent strains, we were able to identify the mutations that were fixed in the populations over the observed time period, giving insight into the evolution of these bacteria. These results indicate that bacterial strains can persist in the vagina for extended periods of time, providing an opportunity for them to evolve in the host microenvironment.

**IMPORTANCE** The stability of strains within the vaginal microbiota is largely uncharacterized. Should these strains be capable of persisting for extended periods of time, they could evolve within their host in response to selective pressures exerted by the host or by other members of the community. Here, we present preliminary findings demonstrating that bacterial strains can persist in the vagina for at least 1 year. We further characterized *in vivo* evolution of the persistent strains. Several participants were also found to not have persistent strains, despite having a vaginal microbiota (VMB) with similar species composition at the two time points. Our observations motivate future studies that collect samples from more participants, at more time points, and over even longer periods of time. Understanding which strains persist, what factors drive their persistence, and what selective pressures they face will inform the development and delivery of rationally designed live biotherapeutics for the vagina.

**KEYWORDS** vaginal microbiome, evolution, metagenomics, temporal stability

Address correspondence to Jacques Ravel, jravel@som.umaryland.edu.

The authors declare a conflict of interest. J.R. is a cofounder of LUCA Biologics, a biotechnology company focusing on translating microbiome research into live biotherapeutic drugs for women's health. All other authors declare no competing interests.

The human microbiome is estimated to be comprised of hundreds to thousands of distinct bacterial species and strains (1, 2). These bacteria often live in close association with host tissues and are thought to be critical for the maintenance of our health (3). Studies on the gut microbiome have demonstrated that strains of these species are capable of persisting within a host for extended periods of time (4, 5). However, the

potential for long-term persistence of bacterial strains at other body sites remains largely unexplored. The microbial communities that inhabit the vagina are unique from those found at other body sites (6). They are often dominated (>90% relative abundance) by single species of *Lactobacillus*, although a significant proportion of women, around a third, have more compositionally even communities containing an assortment of facultative and obligate anaerobic bacteria (7, 8). Communities that are dominated by *Lactobacillus* spp. have been associated with a decreased risk for several adverse health outcomes, leading many to consider them to be "optimal" (reviewed by France et al. [9]). Observational studies have demonstrated that the vaginal microbiota (VMB) of some women maintain species composition over several menstrual cycles, while others have communities that vary over time (10, 11). A small study on the VMB of pregnant women found results consistent with the persistence of strains throughout gestation (12). It has yet to be determined whether the VMB typically maintains bacterial strain composition over longer periods of time or whether there are frequent turnovers in the dominant strain of each species.

In this study, we sought to characterize the long-term persistence and *in vivo* evolution of bacterial strains within the VMB. Women whose communities had similar species composition at time points separated by at least 1 year were identified using previously published 16S rRNA gene amplicon survey data (7, 11), and 10 were selected to represent the breadth of commonly observed community compositions. The protocol was approved by the Institutional Review Boards of Emory University School of Medicine, Grady Memorial Hospital, and the University of Maryland School of Medicine. Written informed consent was appropriately obtained from all participants that included permission to use the samples obtained in future studies.

Shotgun metagenomes were generated to characterize the strains present at each time point (13). The resulting sequence reads were mapped to the VIRGO nonredundant gene catalog (14) to establish the taxonomic composition of each sample (Fig. 1A, complete methods in Text S1). All participants had similar species in their VMB at the two time points, but in the cases of participants 4 and 5, their relative abundances had shifted substantially. Both of these participants had communities that were dominated by *Lactobacillus iners* at the initial time point, which was later supplanted by either *Lactobacillus crispatus* (participant 4) or *Lactobacillus jensenii* (participant 5).

We next sought to determine which participants had maintained the same strain(s) over the 1- to 2-year time period. *De novo* assembly using metaSPAdes (15) and contig binning, as described previously (16), produced 53 metagenome assembled genomes (MAGs), representing 15 species (Fig. 1B). To identify which participants had the same strain(s) at the two time points, we used inStrain (17), with a percent identify threshold of at least 99.999%. This relatively strict threshold was chosen as any greater degree of sequence divergence would be difficult to explain given estimated substitution rates for bacteria (18). Ten strains representing nine species were identified at both time points from a single individual (Fig. 1C). We conclude that these observations result from the persistence of the strain(s) within a participant's VMB. Of the ten participants, six were found to have at least one persistent strain in their VMB, and three (participants 5, 8, and 9) had multiple persistent strains (Fig. 1A). Of those, two (participants 5 and 8) had communities that were primarily comprised of two species whose strains had persisted (participant 5, *L. iners* and *L. jensenii*; participant 8, *B. longum* and *L. gasseri*). Participant 9 had a more diverse VMB that was not dominated by *Lactobacillus* spp. and was found to have three persistent strains, including two strains of *Prevotella* and one strain of *Ca. L. vaginae*. This observation indicates that strain persistence is not just a property of *Lactobacillus* dominant communities and that these more compositionally even communities can also exhibit long-term stability in strain composition. An additional six strains (four *Gardnerella*, one *L. iners*, and one *Megasphaera*) were estimated to have an average nucleotide identity (ANI) between 99.95 and 99.99%. They could represent persistent strains whose sequence divergence has been inflated either due to recombination or as a result of the low and variable sequence coverage of these

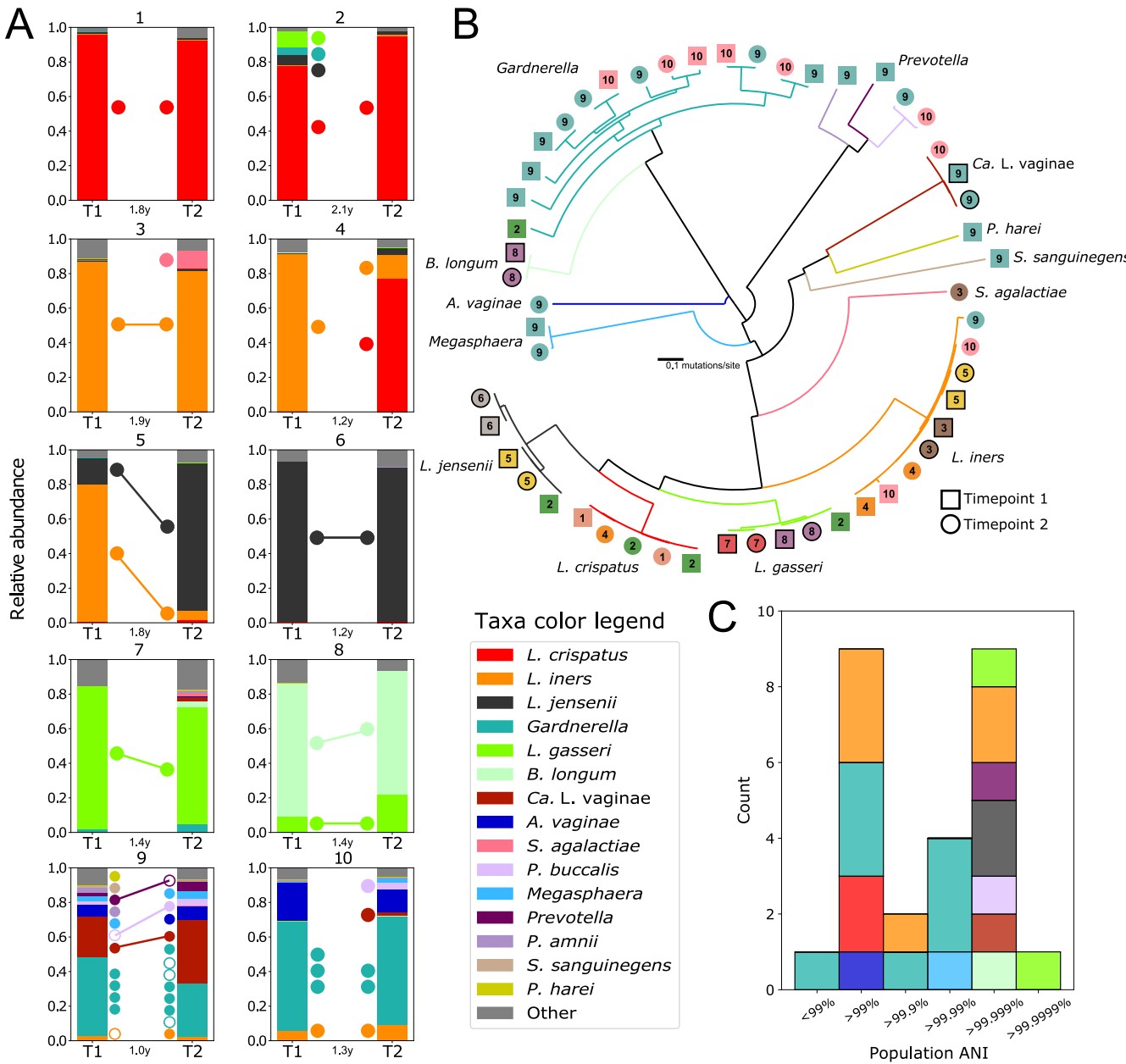

**FIG 1** (A) Taxonomic composition of each participant's vaginal microbiota (VMB) at the two time points as established using the VIRGO nonredundant gene catalog. The points between the bars represent the metagenome assembled genomes (MAGs) recovered from each time point (closed) or strains identified by inStrain but not assembled (open). Where the two points (closed or open) are connected by a line, the strain met the 99.999% ANI threshold and was considered to be present at both time points. (B) Phylogeny of identified MAGs. The phylogenetic tree was derived from a concatenated alignment of 100 orthologous genes identified in at least 98% of the 53 bacterial metagenome assembled genomes. Branches are colored according to taxonomy, and MAGs are labeled with the participant number and the time point (square, time point 1 [T1]; circle, time point 2 [T2]). MAGs identified as the same strain by the inStrain analysis are outlined in black. (C) Average nucleotide identity (ANI) between strains of the same species that were identified at both time points for a participant. Stacked bars are colored according to taxonomy.

species. Conversely, these observations could represent instances in which a closely related strain of the same species displaced the original.

The remaining four participants were not found to have any persistent strains, despite the similarity in their taxonomic composition at the two time points. This included the two participants that had a *L. crispatus* dominant VMB at both time points (participants 1 and 2, Fig. 1A), indicating that the *L. crispatus* at the initial time point was supplanted by another at the second sampling. It could be that these *L. crispatus* populations went extinct and were subsequently reestablished or that the population experienced a shift in the dominant

**TABLE 1** Sequence differences identified in persistent strains

| Participant | Species | No. differences[a] | Genes with differences[b] |
|---|---|---|---|
| 3 | *L. iners* | 13 (5/5/0/3) | **wecH**, *DagR*, **sbnD**, **yheH**, *mglA*, **recF** |
| 5 | *L. iners* | 15 (7/1/2/5) | **adk**, **spxA**, *rfbX*, **btuD**, **dhaL** |
| 5 | *L. jensenii* | 10 (6/2/1/1) | **yheL**, *dedA*, **ebh**, **arcD1**, **mutS2** |
| 6 | *L. jensenii* | 13 (3/2/5/3) | *yajL*, *glf*, *citX*, *glvR* |
| 7 | *L. gasseri* | 10 (1/0/3/6) | **nrdD**, **hpt**, **cls** |
| 8 | *B. longum* | 22 (6/4/1/11) | **degA**, *gndA*, **thiC**, *nusB*, **hadl**, **glgB**, *acn* |
| 9 | *Ca. L. vaginae* | 8 (2/5/0/1) | *lsaC* |
| 9 | *Gardnerella gsp7* | 15 (0/11/0/4) | *glgE*, *gatB*, *thiO*, *carA*, *ykoE*, *ybhL* |

[a]Number of differences (nonsynonymous/synonymous/indel/intergenic).
[b]Genes with differences, those in bolded genes caused a change in amino acid sequence, genes encoding
 hypothetical proteins not shown.

strain, as prior studies have indicated these populations are often comprised of multiple strains (14, 19). Participant 4 had a *L. iners*-dominated VMB at the initial time point, which shifted to a community that contained a majority of *L. crispatus* and a minority of *L. iners* at the second time point. The *L. iners* identified at the second time point was not the same strain as that identified at the initial time point. Finally, participant 10 had the more diverse VMB at both time points with similar species composition but, unlike participant 9, was not found to have any persistent strains. These observations demonstrate that consistency in species composition in a VMB over time does not necessarily reflect the persistence of individual strains.

Long-term colonization of a strain in the VMB provides an opportunity for the strain's population to adapt to a specific host environment. For the six participants with persistent strains, we were able to identify the specific sequence differences that had accumulated over the observed time period. BreSeq was used to characterize genomic changes in the eight persistent strains that were found to have sufficient coverage (20). The sequence differences were observed in a variety of genetic loci and included nonsynonymous and synonymous changes, as well as small insertions or deletions (summarized in Table 1; details in Table S1). The average number of sequence differences observed for each strain was ~10. Some of these sequence differences, chiefly those that produce an amino acid change, could be adaptive, although it is impossible to discern without additional evidence. Observations at intervening time points could inform the order and speed of each fixation event and reveal which mutations occurred on the same background and fixed jointly. Additionally, observing the long-term strain persistence and evolution in a larger cohort could reveal instances of parallel evolution, indicating an adaptive role.

In conclusion, we observed the persistence of bacterial strains in the vaginal microbiota over a 1- to 2-year period. This included several instances in which multiple bacterial strains persisted together in the same community. It is not clear why some participants maintained their strains while others did not. Host factors are expected to play a principal role and would include things like the use of antibiotics or the introduction of novel sexual partners (21). However, it could also be that some species or strains are more capable long-term colonizers of the vaginal niche than others and that microbial factors play a disproportionate role. We may have also failed to detect persistent strains due to a lack of sequence depth, and persistence may be more common than suggested by our results. Larger studies are needed to characterize the determinants of strain persistence in the vaginal microbiota. For the persistent strains, we were also able to characterize changes in genomes of the persistent strains. These observations provide an initial glimpse into the *in vivo* evolution of the vaginal microbiota. From these preliminary data, we conclude that the strain composition of the vaginal microbiota is often stable over long periods of time but should not be assumed.

**Data availability.** The shotgun metagenomic data have been deposited in the Short Read Archive: PRJNA575586 (see Table S2 for specific BioSample and SRA accession

numbers). All scripts used in the processing and analyses of the metagenomes are available at: https://github.com/ravel-lab/two_year.

## SUPPLEMENTAL MATERIAL

Supplemental material is available online only.
**TEXT S1**, DOCX file, 0.03 MB.
**TABLE S1**, XLSX file, 0.02 MB.
**TABLE S2**, XLSX file, 0.01 MB.

## ACKNOWLEDGMENTS

This work was supported in part by the National Institute of Allergy and Infectious Diseases and the National Institute for Nursing Research of the National Institutes of Health under awards UH2AI083264 and R01NR015495, respectively.

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
