## [Reviewer comments · mSystems]

Persistence and *in vivo* evolution of vaginal bacterial strains over a multi-year time period

Michael France, Bing Ma, and Jacques Ravel

Corresponding Author(s): Jacques Ravel, University of Maryland School of Medicine

Review Timeline:

Submission Date:	September 14, 2022
Editorial Decision:	October 14, 2022
Revision Received:	October 21, 2022
Accepted:	November 7, 2022

Editor: Nandita Garud

Reviewer(s): The reviewers have opted to remain anonymous.

Transaction Report:

DOI: <https://doi.org/10.1128/msystems.00893-22>

October 14, 2022

Dr. Jacques Ravel
University of Maryland School of Medicine
Institute for Genome Sciences
HSF III - Room 3173
801 W. Baltimore Street
Baltimore, MD 21201

Re: mSystems00893-22 (Persistence and *in vivo* evolution of vaginal bacterial strains over a multi-year time period)

Dear Dr. Jacques Ravel:

Thank you for submitting your manuscript to mSystems. We have completed our review and I am pleased to inform you that, in principle, we expect to accept it for publication in mSystems. However, acceptance will not be final until you have adequately addressed the reviewer comments.

We would be happy to extend an accept if the following two modifications can be made:

1. Please adhere to a 99.999% ANI threshold as suggested by Reviewer 1. Reviewer 1 thinks that this should not change the conclusions substantially, and is a simple enough fix to address any potential confusion with the definition of persistence.
2. Please address comments from reviewer 2. "The authors might choose to qualify that the absence of detected shared strains might in some case be limited by data quality and the limits of inStrain, and that persistence might be even more common than suggested by these results."

Preparing Revision Guidelines

Sincerely,

Nandita Garud

Editor, mSystems

Journals Department
Reviewer comments:

Reviewer #1 (Comments for the Author):

Thank you for the revisions to improve the manuscript.

I maintain that strains with <99.999% ANI are not necessarily "likely persistent," any more than they were "persistent" in the previous version. Especially if the data suffer from "low and variable sequence coverage," it is difficult to know whether recombination, for example, happened within a person over the course of the study or whether another strain, with a history of recombination elsewhere, migrated in to replace the original strain. Both persistence and replacement are observed in this data set. I am not convinced that persistence has a strong enough prior to justify this interpretation.

Reviewer #2 (Comments for the Author):

Thank you to the authors for addressing my comments. The authors have sufficiently addressed all my questions and I am satisfied with these revisions.

The authors might choose to qualify that the absence of detected shared strains might in some case be limited by data quality and the limits of inStrain, and that persistence might be even more common than suggested by these results.

Response to Reviewer comment:

(responses appear in blue)

Reviewer #1 (Comments for the Author):

Please adhere to a 99.999% ANI threshold as suggested by Reviewer 1.

We have removed the dotted lines from Figure 1 indicating likely persistence and have similarly revised the text which now reports these strains and their ANI but does not conclude that they have persisted.

P4 L 102-107: “An additional six strains (four *Gardnerella*, one *L. iners*, and one *Megasphaera*) were estimated to have an ANI between 99.95 and 99.99% ANI. These could represent persistent strains whose sequence divergence has been inflated either due to recombination or as a result of the low and variable sequence coverage of these species. Conversely, these observations could represent instances where a closely related strain of the same species displaced the original.”

Reviewer #2 (Comments for the Author):

The authors might choose to qualify that the absence of detected shared strains might in some cases be limited by data quality and the limits of inStrain and that persistence might be even more common than suggested by these results.

We have revised the text as suggested to qualify the limitations of our analysis.

P5 L 148-150: “We may have also failed to detect persistent strains due to a lack of sequence depth, and persistence may be more common than suggested by our results.”

November 7, 2022

Dr. Jacques Ravel
University of Maryland School of Medicine
Institute for Genome Sciences
HSF III - Room 3173
801 W. Baltimore Street
Baltimore, MD 21201

Re: mSystems00893-22R1 (Persistence and *in vivo* evolution of vaginal bacterial strains over a multi-year time period)

Dear Dr. Jacques Ravel:

Your manuscript has been accepted, and I am forwarding it to the ASM Journals Department for publication. For your reference, ASM Journals' address is given below. Before it can be scheduled for publication, your manuscript will be checked by the mSystems production staff to make sure that all elements meet the technical requirements for publication. They will contact you if anything needs to be revised before copyediting and production can begin. Otherwise, you will be notified when your proofs are ready to be viewed.

Publication Fees:

If you would like to submit a potential Featured Image, please email a file and a short legend to mSystems@asmusa.org. Please note that we can only consider images that (i) the authors created or own and (ii) have not been previously published. By submitting, you agree that the image can be used under the same terms as the published article. File requirements: square dimensions (4" x 4"), 300 dpi resolution, RGB colorspace, TIF file format.

We recognize that the video files can become quite large, and so to avoid quality loss ASM suggests sending the video file via <https://www.wetransfer.com/>. When you have a final version of the video and the still ready to share, please send it to mSystems staff at mSystems@asmusa.org.

Sincerely,

Nandita Garud
Editor, mSystems

Journals Department
Supplemental Table 2: Accept
Supplemental Text 1: Accept
Supplemental Table 1: Accept